

# ESD Ideas: Structures dominate the functioning of Earth systems, but their dynamics are not well represented

Axel Kleidon[1], Erwin Zehe[1], Ralf Loritz[2]

[1]Max Planck Institut für Biogeochemie, Jena, Germany
[2]Karlsruhe Institute of Technology (KIT), Institute of Water and River Basin Management, Karlsruhe, Germany

*Correspondence to*: Axel Kleidon (akleidon@bgc-jena.mpg.de)

**Abstract.** Many fluxes in Earth systems are not homogeneously distributed across space, but occur highly concentrated in structures, such as turbulent eddies, river networks, vascular networks of plants, or human-made infrastructures. Yet, the highly-organized nature of these fluxes is typically only described at a rudimentary level, if at all. We propose that it
requires a novel approach to describe these structures that focuses on the work done to build and maintain these structures, and the feedbacks that they cause on a system's ability to perform work, which requires placing these structures into their environmental Earth system context.

## 1 Structures in Earth systems

At the very core of many physical models of Earth system components are energy, mass, and momentum balances,
representing physical conservation laws. The implementation of these physical laws then result in the dynamics described by changes in storage terms and fluxes. Examples include the surface energy balance, with radiative and heat fluxes determining the dynamics of surface temperature, the soil water balance, describing the dynamics of soil moisture in relation to the fluxes of infiltration, evapotranspiration and runoff, or the momentum balance in the atmosphere, which in form of the Navier-Stokes equation sets the basis for the dynamical cores of atmospheric models. These formulations are typically
represented at a discrete grid and at temporal resolution, with finer grids and smaller time steps resulting in numerically more intensive simulation models.

Yet, when we look at nature, we notice that systems often do not operate at a rectangular grid scale, but form highly organized structures in which the flows are concentrated in. Figure 1a shows one example of a drainage structure that
temporarily formed after a heavy rainfall event at a beach. Being formed at a scale of less than a meter, it is typically not being resolved in grid-based simulation models. Yet, it is very likely that the structure has, locally, a strong impact on the flow, allowing for faster drainage than in the absence of the structure. This acceleration of the flow is a consequence of lines of research that attribute the formation of such network structures to the minimization of frictional dissipation (e.g., Howard 1990, Rinaldo et al. 1992, Rodriguez-Iturbe and Rinaldo, 1997 for river structures, West et al. 1997 for vascular tree



networks, and West, 2017 for further examples).  When included with the surrounding driving gradient, it appears that such structures act to deplete driving gradients at a faster, possibly even maximized rates (Kleidon et al. 2013).

The prevalence of such structures is not restricted to drainage structures.  Other examples of such flow structures include turbulent structures in air and water flow, vascular networks of plants that efficiently conduct water from the soil to the

canopy, and human-made infrastructure such as sewage systems, trade routes, or electrical power grids.  These structures have their own dynamics.  The goal here is to outline that such structures should share the same general dynamical foundation, and this includes the effects that these structures have on their environmental system.

## 2. Structures require work

We argue that the starting point to represent structures is to recognize that these require physical work to be built and to be

maintained, e.g., to accelerate air into circular motion or to detach sediments to shape channels.  The source of this work, however, can differ depending on what type of structure we deal with.  This requirement for work links the structure to its Earth system context as this is where the ability to perform work comes from.  Furthermore, we argue that the effect that structures have on the associated flows within the system is such that it affects the ability to perform this work.

We propose that the dynamics of structures can be represented as a linear, differential equation of the form

$$\frac{dU_{structure}}{dt} = u_{structure} \cdot \frac{dA}{dt} + A \cdot \frac{du_{structure}}{dt} = G - D \ . \tag{1}$$

Here, $U_{structure}$ represents the past work done to build the structure (in units of Joule, with $u_{structure}$ being the work done per unit area and $A$ being the spatial extent of the structure), $G$ represents the power (work/time, in units of Watt) to build and maintain it, and $D$ is the decay of the structure due to dissipative processes that come at an energetic cost to the system (in

units of Watt).  To first approximation, we can assume this decay to be proportional to $U_{structure}$, with a typical time scale $\tau$ representing the proportionality.  This leads to $D = U_{structure}/\tau$.  Eq. (1) thus represents a relatively simple differential equation for a specific type of work that is represented by the structure.

We can now classify structures in terms of the type of work that they represent, the power source, the lifetime of the

structure, and which Earth system processes are affected.  This is illustrated using the following four examples:

**Turbulent structures:** Turbulent structures are represented by turbulent kinetic energy that is generated out of buoyancy work due to heating or cooling, or out of the shear stress from a mean flow. The time scale is typically short as these structures form at time scales of seconds to minutes, although large-scale turbulent structures, such as hurricanes or



atmospheric pressure systems may take days to form. Their presence dominates the transport of momentum, heat and moisture, so they play a critical role, particularly for surface-atmosphere exchange and large-scale heat transport.

**Drainage structures:** These hydrologic structures are represented by work done by relocating sediments to form rills and channels. The work comes from the shear stress of the water flowing over the surface. The time scale of their formation

spans minutes (like the drainage structure in Fig. 1a formed during a rain shower) to centuries, with larger river network structures at the catchment or continental scale being characterized by much longer time scales. The presence of these structures affect the frictional dissipation of water flow as they reduce the contact area of the water to the sediment. The result is that these structures dominate the water and sediment flows on land.

**Plant structures:** Vascular structures are made of biomass, or chemical energy, with the energy source being photosynthesis. Their lifetime spans from weeks to months for annual plants to decades and centuries for trees. These structures are crucial in supplying leaves with the water they need for transpiration and for taking up carbon dioxide. The continuation of the vascular structures into the veins of leaves is a critical bottleneck for achieving high photosynthetic rates (Boyce et al., 2009). It would thus seem that these structures are not only highly efficient in transporting water to the

canopy, but that this is required to achieve higher photosynthetic rates and thus generating the energy from which these structures are made from.

**Human-made structures:** There are various, man-made structures, and the energy used to build and maintain them typically comes either from the physical work done by humans or livestock (for pre-industrial structures) or from a primary energy

source such as fossil fuels (for industrial structures). While it would require more detail to formulate their individual roles, it would nevertheless appear that they follow the same characteristic patterns: They typically make flows more efficient (such as distribution networks for drinking water, sewage, trade, or electricity), thus allowing for more human activity.

**3. Stages of development**

We can see immediately from Eq. (1) that these very different types of structures should share general developmental stages

as they grow, develop, and increasingly affect the flows of their environment. These stages are characterized by the relative magnitudes of $G$ and $D$, but also by the feedbacks of the structure (as characterized by $U_{structure}$) on the environment from which the work is derived from (as outlined in Fig. 1b).

We can characterize the development into a minimum set of five stages:




a. An **initial growth stage** is characterized by $G > 0$ and, while the structure is still small ($U_{structure} \approx 0$), $D \approx 0$. It describes a period in which the structure grows (as $G > D$), and so does the associated work to build the structure and increase $U_{structure}$. As the structure is small, its effects on the environment is quite small.

b. As the structure grows in extent, it enters a **positive feedback stage.** As the structure reduces the frictional losses of the flows in its immediate environment, this results in a positive feedback as more work can be derived from the flow to grow and develop the structure.

c. With further growth and reduced frictional losses, the flows that are associated with the structure dominate and
increasingly deplete their driving gradient in the environment. This results in a **negative feedback stage** as the driving gradient is depleted faster and the rate of net growth, $G - D$, slows down.

d. Eventually, the structure reaches a **maintenance stage** of a quasi steady-state when the averaged power balances its depletion, so $G = D$ in the mean. In this stage, both, the positive and negative feedbacks have equal strengths, but
are of opposite sign. The structure no longer grows in size.

e. The structure can enter a **decay stage** when $G < D$, so that the work put into the structure is less than its decay, for instance, when the environment changes and reduces the flow from which the work is derived. This implies that the structure reduces in its extent and deteriorates, including its effects on the environment.

The time scale involved in passing through these stages links to the time scale $\tau$ described above. This can be directly seen by the mean steady state of Eq. (1) in which $G = D$, which yields $\tau = U_{structure}/G$. The time scale thus depends on the power $G$ involved in building the structure as well as the work needed to build a structure of a certain extent ($U_{structure}$), but also on the effect that the resulting structure has on the flows and the associated feedbacks.

An example for structures with a short timescale of formation are turbulent structures associated with land-atmosphere exchange. They are formed out of comparatively large power $G$ due to the strong heating by absorption of solar radiation, and they require a comparatively low $U_{structure}$ (i.e., it requires less work to build structures in air than in solids) to have an effect on transporting heat from the surface to the atmosphere. The maintenance stage is thus reached quickly. This is
probably the reason why turbulent exchange is well described by thermodynamic limits, such as the maximum power limit, as these set the limit to how much power can maximally be derived from the environmental setting (e.g., Ozawa et al., 2003; Kleidon, 2016).



An example for structures with a long timescale are river networks. They are associated with much less power and require much more work to be built as it requires the relocation of sediments. They develop on much longer time scales than the turbulent structures in the atmosphere. These structures may thus not have reached their thermodynamic limit for a given environment. It would nevertheless seem that thermodynamics would then set a general evolutionary direction towards this limit for the development of the structure towards stage (d.) at which power and dissipation are maximized.

## 4. Challenges and conclusions

To develop this explicit description of structures and their role in Earth systems further would require progress in developing a physical Earth system perspective that does not just represent conservation laws, but focuses on the thermodynamics, as it sets the basic rules and limits for deriving work. While here we focused on the work done to build structures, a more general basis of how Earth system processes perform work is still poorly developed. This basis, however, sets the foundation for determining where structures get their work from and how structures affect their environment. Furthermore, it would require a shift away from the grid scale to the scale at which structures form and develop. This, in turn, may happen well within a typical grid cell, but it may also span many grid cells, as in the case of river networks.

A more explicit description of the dynamics of structures would have substantial potential to advance our understanding of how and why these structures form along with their associated scaling laws, and how these relate to general thermodynamic evolutionary trends and optimality principles such as maximum power, minimum dissipation, or maximum entropy production. At the more applied level, it should yield a better understanding how and how fast systems are able to adapt to change, thus improving our ability to understand the impacts of global climate change on Earth system processes and their organization.

**Acknowledgements.** This research contributes to the "Catchments As Organized Systems (CAOS)" research group (FOR 1598) funded by the German Science Foundation (DFG).

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

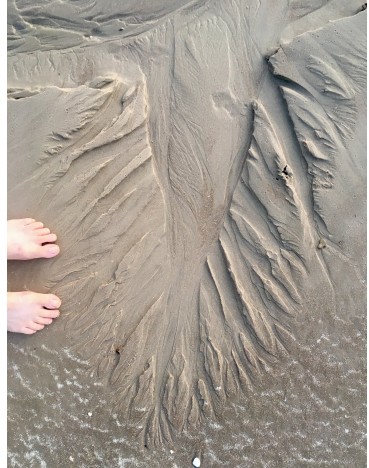

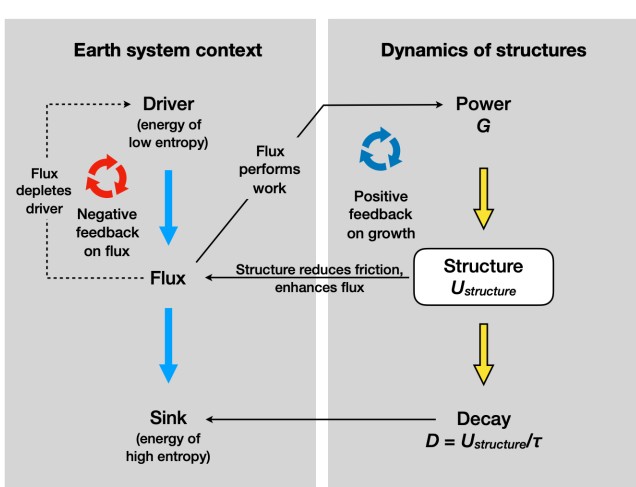

**Figure 1. (left) An example of a small drainage structure that formed at a beach during a rain shower. (right) Schematic diagram to illustrate how the dynamics of structure growth and decay link to environmental fluxes and associated feedbacks.**
