# Peer review of "ESD Ideas: Structures dominate the functioning of Earth systems, but their dynamics are not well represented"

_Earth System Dynamics, 2019_

## Referee Comment (RC1) · Ralph Lorenz (Referee) · 25 Nov 2019

This paper raises the valuable concept of structures in systems (Earth, societal, planetary etc.) and notes that the growth and sustainment of structures should be able to be quantified. This seems like a promising avenue of enquiry, but the present paper fails to adequately develop the concept. While I recognize that an 'ESD ideas' format should encourage advancement of concepts that may not be fully mature, the present very short manuscript really doesn't do enough by itself to merit publication. In particular, there is a yawning gap – not to say a gross inconsistency in physical dimensions/units – between the (obvious, and qualitatively well-discussed in previous

works – e.g. Lineweaver, Bejan and others) structures like vascular networks, sewage systems and flow structures (lines 32-37) and the \*\*energies\*\* discussed in equation 1 in lines 45-52. For each of the environments discussed in section 2, the authors should make at least a token effort to (1) define the physical quantities involved (kinetic energy and viscous dissipation in flow, volume of material and transport rates in geomorphic structures like rivers, metabolic rate and biomass? in plants etc.), and (2) identify the destruction mechanisms against which growth must compete – otherwise the paradigm is meaningless. Then, for at least a couple of these, provide a numerical example or two where these properties, and the resultant timescale, is actually quantified. This exercise, which probably involves half an afternoon, some coffee and a whiteboard, could turn this half-developed 'placeholder' of analogies into a valuable contribution to the literature where the idea is shown to have predictive utility.

---

## Referee Comment (RC2) · Steffen Birk (Referee) · 3 Dec 2019

This paper presents the idea to describe the dynamics of flow structures by considering the work required to build and maintain the structures. This is an interesting idea, and I agree that it could be particularly attractive to explore its relation to optimality principles as suggested at the end of the paper. However, I wonder if the different types of structures discussed by the authors actually fit into one approach or at least whether it is helpful to address such different phenomena by one approach. It seems to me that the meaning of the term "structure" is quite different in the given examples, e.g. with regard to the source of the energy, which in some cases (e.g. drainage structures) may be

related to the flow itself (thus potentially leading to self-organized flow patterns), while it is external in other cases (plants, human-mad structures – the degree of coupling between the external source of energy and the flow structure appears to be very different in these cases). Thus, I wonder if the development stages proposed by the authors are actually applicable to all of these structures. In summary, I think the concept needs further thought and explanation.

Specific comments:

1) Line 47: Is it actually "the work done to build the structure"? As I understand, the energetic costs due to dissipative processes are subtracted from the work done to build the structure. Thus, U seems to represent the work that would be needed to build the structure in the absence of decay.

2) Lines 34 and 56: I wonder if a "turbulent structure" is a structure in the sense of the structure of e.g. river channels. Flow in the river channel also can be turbulent, but still the channel itself is considered to be the structure (whereas turbulence would be considered a state of the flow). Unlike a river channel or plant structures, turbulence does not appear to meet the requirement (line 34) that it "efficiently " conducts water, as laminar flow is more efficient (in that frictional losses are lower). Examples of structures that appear to be similar to river channels are preferential pathways in soils or solution conduits in carbonate (or gypsum) aquifers (see Hergarten et al., Hydrol. Earth Syst. Sci. , 2014, doi:10.5194/hess-18-4277-2014). With regard to the latter, it is interesting that Howard and Groves (Water Resources Research, 1995, doi:10.1029/94WR01964) found highly selective development of solution conduits under laminar flow, whereas "the transition to turbulent flow results in more general passage enlargement ", i.e. turbulence does not seem to favor flow concentration in localized structures. I guess convection cells, jet stream etc. might be considered structures (i.e. the corresponding pressure distribution resulting from temperature/density differences) that efficiently conduct atmospheric flows.

3) Lines 95-97: "...positive feedback ... more work can be derived from the flow to grow and develop the structure" – this applies e.g. to the river channels and also to the above mentioned solution conduits, where the flow creates the structure. I am not sure if it applies to plant structures – this might seem reasonable if plant growth is water limited but what if growth is e.g. energy limited, why should there be a feedback with flow? Even more so in the case of human-made structures, why would "more work derived from the flow" enhance the development of the structure? Whether a human-made structure is further developed, maintained or decays will likely depend on socioeconomic conditions or more generally the development of the human-environment system. Perhaps there are cases where the feedback of the structure on other components of the human-environment system is sufficient to create a feedback loop, but it seems unlikely that this can be generalized.

4) Lines 98-101: Similarly, this may not apply to all structures considered here.

5) Line 128 – "dissipation ... maximized" vs. line 140 "minimum dissipation": Line 128 suggests that the structure develops to a stage where dissipation is maximized, whereas line 140 suggests that "a more explicit description of the dynamics of structures would ... advance our understanding ... how they relate to optimality principles such as ... minimum energy dissipation" – this might appear contradictory and thus it should be further explained how the development of the structure towards maximum dissipation is consistent with the concept of minimum dissipation. Is it because the first refers to the localized structure only, whereas the second refers to the entire system? If so, again (as discussed in comment no. 3) the relationship between the structure and the entire system (catchment, plant, human-environment system, etc.) needs to be addressed, in particular, the feedbacks between the two, which might be very different for the various structures mentioned in the paper.

---

## Author Comment (AC1) · 29 Jul 2020

**Review #1 by Ralph Lorenz:**

We thank Ralph Lorenz for his constructive comments. In the following, we separated his review into five comments (in **bold**, numbered) and provide a response to each of them, followed by how we plan to address this comment in the revision.

***Comment R1-1. This paper raises the valuable concept of structures in systems (Earth, societal, planetary etc.) and notes that the growth and sustainment of structures should be able to be quantified. This seems like a promising avenue of enquiry, but the present paper fails to adequately develop the concept. While I recognize that an 'ESD ideas' format should encourage advancement of concepts that may not be fully mature, the present very short manuscript really doesn't do enough by itself to merit publication.***

*Response:* Yes, the manuscript is very short -- this is one of the requirements of the ESD Ideas format. We also agree that the concept we propose is not fully mature, which is why we chose this manuscript type. As we will mention in the following, there is a huge gap in Earth system science in terms of formulating structures as the result of work being done. Most processes in the Earth system are not even described in terms of how much work is being done, so it is, obviously, difficult to get some numbers to illustrate this point. It illustrates a huge lack of the basics to even describe such structures in energetic terms.

In our manuscript, we aim to provide the concept to advance towards such a description of structures. We think this fits very well into the category of *ESD Ideas*, because it is novel and innovative. The reviewer makes several constructive remarks that we respond to in the following that will hopefully make the manuscript stronger.

*Action:* In the revision, we will describe better the challenging context of Earth system processes not being formulated in terms of how much work these perform. We will also aim to strengthen the manuscript by addressing the other points raised by the reviewer as detailed below.

***Comment R1-2. In particular, there is a yawning gap – not to say a gross inconsistency in physical dimensions/units – between the (obvious, and qualitatively well-discussed in previous works – e.g. Lineweaver, Bejan and others) structures like vascular networks, sewage systems and flow structures (lines 32-37) and the \*\*energies\*\* discussed in equation 1 in lines 45-52.***

*Response:* We think the reviewer errs here. We could not identify an inconsistency in units of the terms of Equation 1 and the following description. Equation 1 is given by

$$\frac{dU_{structure}}{dt} = u_{structure} \cdot \frac{dA}{dt} + A \cdot \frac{du_{structure}}{dt} = G - D \tag{1}$$

The individual terms of the equation and their respective units are:

| Term | Description | Unit |
|---|---|---|
| $U_{structure}$ | Net work done to build the structure | J |
| $dU_{structure}/dt$ | Temporal change in $U_{structure}$ | J/s or W |
| $u_{structure}$ | Net work done per unit area, $U_{structure}/A$ | J/m$^2$ |
| $A$ | Spatial extent of structure | m$^2$ |
| $dA/dt$ | Change in spatial extent | m$^2$/s |
| $G$ | Power (work done per unit time) | J/s or W |
| $D$ | Decay of structure | J/s or W |

The terms in Eq. 1 have thus the following units

$$\frac{dU_{structure}}{dt} \left[\frac{J}{s}\right] = u_{structure} \left[\frac{J}{m^2}\right] \cdot \frac{dA}{dt} \left[\frac{m^2}{s}\right] + A\left[m^2\right] \cdot \frac{du_{structure}}{dt} \left[\frac{J}{m^2 s}\right] = G\left[\frac{J}{s}\right] - D\left[\frac{J}{s}\right]$$

All terms are in units of J/s, or Watt.

One shortcoming of this formulation, which we noticed since submission, is that some structures, such as turbulence in the atmosphere, do not cover areas, but volumes. In this case, the area $A$ would need to be replaced by a volume $V$ instead, with $u_{structure}$ being defined as "net work done per unit volume, $U_{structure}/V$". Yet, the underlying formulation and its interpretation would not be affected.

*Action:* We will clarify the units of Eq. (1) in the revision to make sure that readers can easily see that the units of the equation are consistent. We will also include the last point about volumes for some structures.

***Comment R1-3. For each of the environments discussed in section 2, the authors should make at least a token effort to (1) define the physical quantities involved (kinetic energy and viscous dissipation in flow, volume of material and transport rates in geomorphic structures like rivers, metabolic rate and biomass? in plants etc.), …***

*Response:* We would like to point out that the text in section 2 already contains references to some of the physical quantities involved, yet at a condensed level to keep the text short. For turbulent structures, the reference is made to turbulent kinetic energy, a well-defined quantity in meteorology (see e.g., Peixoto and Oort, Physics of Climate, or Stull, An introduction to boundary layer meteorology).

For drainage structures, we agree that we do not mention the forms of energy involved. It involves work done in terms of detachment of sediment and keeping sediment in suspension when it is transported and relocated by water flow. For plant structures, these are being built by plant processes that are driven by the chemical energy generated by photosynthesis. For human structures, it depends which specific structure is being discussed.

*Action:* We will add text and clarify which forms of energy and work are involved in the revised manuscript and how these are being reflected in the variables associated with the examples.

***Comment R1-4. … and (2) identify the destruction mechanisms against which growth must compete – otherwise the paradigm is meaningless.***

*Response:* The suggestion to identify the destruction mechanism is an interesting point. While this is relatively clear for turbulent structures (molecular diffusion) as well as drainage structures (erosion), for plant structures and human-made structures the destruction processes (such as cell death, plant mortality) are more complicated and less physical.

*Action:* We will mention these destruction mechanisms in the revised version.

***Comment R1-5. Then, for at least a couple of these, provide a numerical example or two where these properties, and the resultant timescale, is actually quantified. This exercise, which probably involves half an afternoon, some coffee and a whiteboard, could turn this half-developed 'placeholder' of analogies into a valuable contribution to the literature where the idea is shown to have predictive utility.***

***Response:*** We agree that having numerical examples are useful to illustrate this concept. Yet, the challenge is that many Earth system processes are not described in terms of how much work these involve, which is part of the motivation for our manuscript.

In atmospheric science, a demonstration of the concept is relatively straightforward. The generation of kinetic energy has already been described in the context of the Lorenz Energy Cycle, an established, yet relatively marginal topic. From estimates of the Lorenz Energy Cycle, the properties and time scales can be illustrated for the case of atmospheric generation and dissipation. Using estimates from the textbook *Physics of Climate* of Peixoto and Oort (1992), we have an estimate of the mean kinetic energy associated with the large-scale atmospheric circulation of $U_{ke} = 7.3 \times 10^5$ J m$^{-2}$ that is associated with the large-scale structures of high and low pressure systems. The generation rate $G$ is about 2 W m$^{-2}$, and this yields a time scale of 7.3 $\times 10^5/2$ s = 4.2 days. This time scale agrees well with the typical time scale of synoptic activity in the mid-latitudes.

For plant structures, one could develop such a time scale from the chemical energy fixed by plants compared to the standing biomass. Because the conversion efficiency of both is the same, the associated time scale is the same as that derived from the carbon balance. Using typical values of 55 GtC/yr of net primary productivity and a standing biomass in vegetation of about 550 GtC, this yields a time scale of 10 years.

For drainage structures, an energetic framework has not been established, but one could possibly derive a time scale from the sediment mass balance, as the sediment flux encapsulates the work done to detach and sustain sediments in suspension. However, to relate it to structures would require to link this work done to the heterogeneity in topography that was generated by the drainage structure, and this, in turn, would require further analysis. At a qualitative level one would nevertheless expect the resulting time scale to be large, because the removed mass associated with river networks is large, while the input of potential energy by rainfall, some of which fuels sediment detachment and transport, is comparatively small.

***Action:*** We will include these examples in the revision to illustrate the time scales.

---

## Author Comment (AC2) · 29 Jul 2020

**Review #2 by Steffen Birk:**

We thank the reviewer for his constructive and stimulating comments.  In the following, we separated his review into one general (in **bold**, not numbered) and five specific comments (in **bold**, numbered) and provide a response to each of them, followed by how we plan to address this comment in the revision.

***General comment. This paper presents the idea to describe the dynamics of flow structures by considering the work required to build and maintain the structures. This is an interesting idea, and I agree that it could be particularly attractive to explore its relation to optimality principles as suggested at the end of the paper.  However, I wonder if the different types of structures discussed by the authors actually fit into one approach or at least whether it is helpful to address such different phenomena by one approach. It seems to me that the meaning of the term "structure" is quite different in the given examples, e.g. with regard to the source of the energy, which in some cases (e.g. drainage structures) may be related to the flow itself (thus potentially leading to self-organized flow patterns), while it is external in other cases (plants, human-made structures – the degree of coupling between the external source of energy and the flow structure appears to be very different in these cases). Thus, I wonder if the development stages proposed by the authors are actually applicable to all of these structures. In summary, I think the concept needs further thought and explanation.***

*Response:* Thank you, this is a very good and stimulating comment.  We agree that the energy source differs between the examples, and also how direct the feedback is between the structure and the flux it affects, and that the manuscript is not clear on this distinction.

Yet, to get started in describing the dynamics of structures, we think that it is more central to focus first on the commonalities, and this is that they all need work to be done to get built.  While this may seem trivial, it is not, because "work", as a well-established physical concept, is literally absent in Earth system science, except for a few, relatively marginal applications.  This work to build structures needs to come from somewhere within the Earth system, linking structures to the processes that build and grow them, and structures have an effect, feeding back to the system by modifying the flows.  This template, and the associated temporal evolution should hold for structures in general, and, as the reviewer writes, should link to thermodynamically-related optimality principles.

In terms of the differences, yes, we agree that the structures we used as examples differ in terms of where the energy comes from.  For turbulent structures in air or water flow, this energy is directly drawn from the kinetic energy of the mean flow.  For drainage structures, the work done to move solids is provided by the kinetic energy of the water flow, so the link is more indirect.  For plant structures, the energy comes from photosynthesis (i.e., from radiant energy to chemical energy), with the link being even more indirect.  Yet, the structures all have their feedbacks, although they become more indirect for the three examples.  We agree that we did not clearly explain these differences in the manuscript.

Nevertheless, what we propose in the manuscript is that such a view on structures is needed to fully describe their role in the Earth system, for which we think that the format of an *ESD Ideas* manuscript is well suited (which, according to the ESD website, "*presents innovative and well-founded scientific ideas in a concise way (no more than two pages, including one figure or table) that have not been comprehensively explored*").  Our goal is not to describe a finalized solution to fit all cases, but to initiate a conceptual advance in bringing the dynamics of structures and their roles to Earth systems science.

*Action:* In the revision we will more clearly define what we mean by structures (strong heterogeneity of the flows) and clarify the commonalities and differences between these.  We will also aim to clarify our goal as this seems to not have been communicated well enough.

**Specific comments:**

***Comment R2-1. Line 47: Is it actually "the work done to build the structure"? As I understand, the energetic costs due to dissipative processes are subtracted from the work done to build the structure. Thus, U seems to represent the work that would be needed to build the structure in the absence of decay.***

*Response:* Yes, agreed.

*Action:* We will adjust the text accordingly.

***Comment R2-2. Lines 34 and 56: I wonder if a "turbulent structure" is a structure in the sense of the structure of e.g. river channels. Flow in the river channel also can be turbulent, but still the channel itself is considered to be the structure (whereas turbulence would be considered a state of the flow). Unlike a river channel or plant structures, turbulence does not appear to meet the requirement (line 34) that it "efficiently" conducts water, as laminar flow is more efficient (in that frictional losses are lower). Examples of structures that appear to be similar to river channels are preferential pathways in soils or solution conduits in carbonate (or gypsum) aquifers (see Hergarten et al., Hydrol. Earth Syst. Sci. , 2014, doi:10.5194/hess-18-4277-2014). With regard to the latter, it is interesting that Howard and Groves (Water Resources Research, 1995, doi:10.1029/94WR01964) found highly selective development of solution conduits under laminar flow, whereas "the transition to turbulent flow results in more general passage enlargement ", i.e. turbulence does not seem to favor flow concentration in localized structures. I guess convection cells, jet stream etc. might be considered structures (i.e. the corresponding pressure distribution resulting from temperature/density differences) that efficiently conduct atmospheric flows.***

*Response:* Turbulent structures should also fit the framework, but we probably did not describe it well enough.  Turbulent structures, such as eddies and convection cells, concentrate the transport of physical properties, specifically momentum, but also, e.g., heat and moisture in the atmosphere, and it requires work to build turbulent eddies.  In meteorology, this work is referred to as the generation of turbulent kinetic energy, and it is typically generated out of the kinetic energy of the mean flow.  Turbulent structures do not require "building material" as in the case of river networks, but they need to be generated and they dissipate, similar to the more permanent river flow structures.  They act to enhance momentum fluxes, so they dissipate the kinetic energy of the mean flow more effectively than just diffusion in the case of laminar flow.  For the growth of a convective boundary layer over the day, the same stages of development can be identified.

For river networks, yes, there are turbulent structures within the stream that dissipate the kinetic energy of the mean flow.  The arrangement of river networks, with channels having curved wetted perimeters as opposed to sheet flow, reduce and minimize this frictional loss.  It results in flow networks that can sustain sediment transport, while also creating steepened slopes at the hillslopes, which in turn can generate more work to detach sediments and which provides the positive feedback to growth.

*Action:* We will include this more detailed description in the revision to clarify the example of turbulence.  We will be more specific about the similarities and differences of the examples and the use of the term "efficiency", and describe better how they fit into the stages that are described in section 3.

***Comment R2-3. Lines 95-97: "...positive feedback ... more work can be derived from the flow to grow and develop the structure" – this applies e.g. to the river channels and also to the above mentioned solution conduits, where the flow creates the structure. I am not sure if it applies to plant structures – this might seem reasonable if plant growth is water limited but what if growth is e.g. energy limited, why should there be a feedback with flow? Even***

*more so in the case of human-made structures, why would "more work derived from the flow" enhance the development of the structure? Whether a human-made structure is further developed, maintained or decays will likely depend on socioeconomic conditions or more generally the development of the human-environment system. Perhaps there are cases where the feedback of the structure on other components of the human-environment system is sufficient to create a feedback loop, but it seems unlikely that this can be generalized.*

**Response:** Again, we agree with the reviewer in the sense that the examples differ in terms of how direct the feedback is on the flow.

For plants, even in so-called "energy-limited" terrestrial ecosystems, the magnitude of their photosynthetic carbon fixation is strongly tied to transpiration and the gas exchange of carbon dioxide with the atmosphere and hence to transport, rather than directly to the availability of light. This is just as it is the case for the so-called "energy-limited" regime of evaporation, which is actually not energy limited because the surface receives a lot more radiant energy than the energy that goes into evaporation (To illustrate this point: Using the global energy balance estimates of Stephens et al. 2012, Nature Geoscience: Absorbed solar radiation at the surface (165 W m$^{-2}$) + Downwelling longwave radiation (= 346 W m$^{-2}$) yields a supply of 511 W m$^{-2}$ at the surface in the global mean. The latent heat flux (88 W m$^{-2}$) utilizes only 17% of this radiant energy, so energy is hardly limiting). The limiting process is rather the convective transport of the atmosphere, which in turn is limited by the energy input and thermodynamics (e.g., Kleidon and Renner, 2013, HESS, also Kleidon, 2016). In other words, it is a transport constraint, not an energy constraint. The same holds for carbon fixation by plants. This has, for instance, been documented by Boyce et al. (2009, see citation in the manuscript), which showed that the higher leaf vein density in angiosperms relate to their higher photosynthetic capacity. We agree that this interpretation is not well-known and needs to be clarified in the manuscript to understand the general importance of flow structures.

For human-made structures, the feedbacks are less direct, yet they should exist, at least for those structures that are sustained through time. Again, the feedback is less direct, but it is through the effect that these structures have on the socio-economic activity that "decides" whether these structures will grow or persist in time, just as in the case for convection or ecosystems.

**Action:** We will add the clarification along the lines described in the response for the plant structures. As a more in-depth discussion of human structures would need more text, but the manuscript type *ESD Ideas* requires short papers, we will reduce the focus on human-made structures and refer to them in the introduction and conclusion as possible future fields of application.

**Comment R2-4. Lines 98-101: Similarly, this may not apply to all structures considered here.**

**Response:** In general, such a negative feedback should generally exist, as it stops structures from keep growing. We nevertheless agree with the reviewer that the directness of the feedback differs among the examples, which is currently not reflected in this text.

**Action:** We will revise the manuscript to be more specific about the differences in the examples, particularly in this section that describes the different stages.

**Comment R2-5. Line 128 – "dissipation ... maximized" vs. line 140 "minimum dissipation": Line 128 suggests that the structure develops to a stage where dissipation is maximized, whereas line 140 suggests that "a more explicit description of the dynamics of structures would . . . advance our understanding . . . how they relate to optimality principles such as . . . minimum energy dissipation" – this might appear contradictory and thus it should be further explained how the development of the structure towards maximum dissipation is consistent with the concept of minimum dissipation. Is it because the first refers to the**

*localized structure only, whereas the second refers to the entire system? If so, again (as discussed in comment no. 3) the relationship between the structure and the entire system (catchment, plant, human-environment system, etc.) needs to be addressed, in particular, the feedbacks between the two, which might be very different for the various structures mentioned in the paper.*

*Response:* Yes, minimum dissipation is likely applicable locally (within the convective atmosphere, the river network or plant vascular network), while maximization at the scale of the whole system, which includes the gradients at the system boundary. In the case of atmospheric convection, with structures being the convection cells, these cells are likely to minimize frictional dissipation internally within the atmosphere, but maximize it at their system boundary. Likewise, river networks minimize frictional dissipation (or energy expenditure, in the terminology of Rinaldo and Rodriguez-Iturbe). This should allow them to carry more sediments further and create steepened slopes at the hillslopes where the sediments are originally derived from. This should result in enhanced depletion of the topographic differences within the reach of the river network. There is thus an important difference between a local view of minimum dissipation within the system versus a system's view on maximizing it that includes the boundary.

*Action:* We will clarify this section along the lines described in the response and will be more specific in clarifying the differences in feedbacks for the different structures.